# Reviewing the Integrated Design Approach for Augmenting Strength and Toughness at Macro- and Micro-Scale in High-Performance Advanced Composites

**DOI:** 10.3390/ma16175745

**Published:** 2023-08-22

**Authors:** Behzad Sadeghi, Pasquale Daniele Cavaliere

**Affiliations:** Department of Innovation Engineering, University of Salento, Via Per Arnesano, 73100 Lecce, Italy; pasquale.cavaliere@unisalento.it

**Keywords:** metal matrix composite, strengthening mechanisms, toughening mechanisms, heterogeneous architecture, energy dissipation

## Abstract

In response to the growing demand for high-strength and high-toughness materials in industries such as aerospace and automotive, there is a need for metal matrix composites (MMCs) that can simultaneously increase strength and toughness. The mechanical properties of MMCs depend not only on the content of reinforcing elements, but also on the architecture of the composite (shape, size, and spatial distribution). This paper focuses on the design configurations of MMCs, which include both the configurations resulting from the reinforcements and the inherent heterogeneity of the matrix itself. Such high-performance MMCs exhibit excellent mechanical properties, such as high strength, plasticity, and fracture toughness. These properties, which are not present in conventional homogeneous materials, are mainly due to the synergistic effects resulting from the interactions between the internal components, including stress–strain gradients, geometrically necessary dislocations, and unique interfacial behavior. Among them, aluminum matrix composites (AMCs) are of particular importance due to their potential for weight reduction and performance enhancement in aerospace, electronics, and electric vehicles. However, the challenge lies in the inverse relationship between strength and toughness, which hinders the widespread use and large-scale development of MMCs. Composite material design plays a critical role in simultaneously improving strength and toughness. This review examines the advantages of toughness, toughness mechanisms, toughness distribution properties, and structural parameters in the development of composite structures. The development of synthetic composites with homogeneous structural designs inspired by biological composites such as bone offers insights into achieving exceptional strength and toughness in lightweight structures. In addition, understanding fracture behavior and toughness mechanisms in heterogeneous nanostructures is critical to advancing the field of metal matrix composites. The future development direction of architectural composites and the design of the reinforcement and toughness of metal matrix composites based on energy dissipation theory are also proposed. In conclusion, the design of composite architectures holds enormous potential for the development of composites with excellent strength and toughness to meet the requirements of lightweight structures in various industries.

## 1. Introduction

As aerospace and automobile industries progress, the demand for materials with superior strength and toughness increases. However, strength and toughness, two crucial properties governing the performance and failure of metal components, often show an inverse relationship. This means that improving strength can compromise fracture toughness, posing an ongoing challenge for material scientists. One effective approach to enhance metal materials is the creation of composites [1,2], which involves incorporating high-performance particles into the metal matrix. These metal matrix composites (MMCs) offer notable improvements in specific strength, stiffness, wear resistance, and other desired properties [3,4,5,6]. Additionally, they possess advantageous characteristics such as high conductivity, thermal conductivity, damping, wear resistance, and multi-functionality, making them valuable for future applications in metal structural materials [7,8,9,10,11].

In aerospace, electronics, and electric vehicles (EVs), the use of lightweight advanced metals, particularly metal matrix composites (MMCs), offers a strategic advantage due to their light weight, high strength, and high toughness. Particularly in the aerospace industry and spacecraft development, advanced aluminum matrix composites (AAMCs) are important base materials that enable cost-effective, multi-functional solutions while meeting fuel efficiency and sustainability goals. This innovative approach to aerospace engineering provides reliable solutions to current challenges and future design requirements. MMCs possess important properties such as high structural efficiency, excellent wear resistance, and desirable thermal and electrical properties that make them valuable in various industries such as automotive, rail, thermal management, aerospace, industrial, recreational, and infrastructure [12]. Replacing conventional aluminum alloys with high-strength and high-toughness aluminum matrix composites (AMCs) can reduce structural weight by about 20% to 25%, while increasing energy performance by about 15% to 20% [2].

AMCs offer immense potential for aerospace applications that are exposed to extreme environmental conditions, such as low temperatures, changing temperatures, and extreme overloads. In industries such as aerospace and electric vehicles, where weight reduction is critical to minimizing fuel consumption and carbon emissions, AMCs offer an exciting solution. By significantly reducing weight while maintaining structural integrity, AMCs enable manufacturers to meet the challenges of weight reduction, fuel efficiency, reliability, and overall cost reduction. Over the next 5 to 10 years, Europe will prioritize space exploration and the development of new energy vehicles, particularly electric vehicles, while aligning with the goals of the Paris Climate Agreement. Experts in AMC technology, such as those at Alvant [13], emphasize the potential of AAMCs to improve the efficiency and performance of electric motors. The installation of AAMCs can result in an impressive 40% reduction in rotor weight for axial flux electric motors, increasing the potential power-to-inertia ratio of the rotor. In addition, AAMCs exhibit excellent thermal resistance, withstanding temperatures up to 300 °C. This exceptional thermal resistance makes AMCs a more suitable material for various applications such as motors, batteries, energy recovery systems, fans, and flywheels compared to other composite materials [14]. However, the negative strength–toughness ratio is a significant obstacle to the full potential and widespread application of MMCs and limits their large-scale development and critical applications [10,11,12]. Therefore, it is imperative to overcome the trade-off between strength and toughness and achieve compatibility between high strength and toughness in innovative MMCs, which is a crucial scientific challenge in the field of research. The design flexibility of composites can be seen in three main aspects: the selection of reinforcements, the adaptation of interfaces, and the design of the structure [15].

Among these aspects, design configuration stands out as the most complicated and difficult task in composite design, fabrication, and processing. From a microstructural perspective, design configuration offers a promising approach to simultaneously improving the strength and toughness of metallic materials, provided that precise control of the structural elements is achieved. However, it is important to note that the relationship between strength and toughness in metal matrix composites is affected by local stress concentration, deformation mismatch, and the presence of dense, large-angle grain boundaries that favor the formation of micropores [16,17]. Conventional approaches to metal matrix composites focus on the homogenization and recombination of a single-phase, single-scale reinforcement matrix. This approach aims to address problems such as insufficient matrix density, agglomeration of nano-reinforcement, structural damage, and interfacial reactions [18,19]. The concept of “homogeneous composite”, inspired by biological composites, has undergone significant development leading to unique composite configurations [20,21,22]. Natural composites, such as bone, exhibit exceptional strength, toughness, light weight, and self-healing ability due to their composite nature and hierarchical organization [21,22]. However, for synthetic materials, the challenge is to understand the interaction of structural factors in improving strength and toughness, and to develop effective processing methods for the precise control of multi-level structures.

Composite configurations serve as the basis for creating sophisticated materials that transcend the limitations of individual parts. These configurations are strategically designed arrangements of different phases, materials, and microstructures that are integrated to achieve specific mechanical, electrical, thermal, and functional properties. The importance of composite configurations lies in their profound influence on material behavior. They enable the production of high-performance materials that meet the requirements of modern engineering applications. Composite configurations can be adapted to different requirements. By intelligently designing the arrangement of the individual phases, scientists can select the desired properties while mitigating limitations. For example, incorporating reinforcing phases into MMCs can dramatically increase stiffness and strength, enabling the material to withstand demanding mechanical loads. Similarly, the incorporation of thermally conductive phases into polymer-based composites can contribute to good thermal conductivity, which is important for applications such as the electricity industry. In addition, composite configurations allow for the manipulation of mechanical performance characteristics such as toughness, fatigue resistance, and fracture behavior. By combining these methods, researchers can stop crack propagation, deflect crack paths, and improve energy dissipation mechanisms, thereby increasing fracture toughness and overall mechanical integrity. This design freedom also extends to tailoring anisotropic behavior, where material properties are varied in different directions to allow for optimal operation under specific conditions. In addition, composite configurations combine different components into a single material system, allowing the materials to perform multiple functions simultaneously. Beyond performance benefits, composite configurations contribute to sustainable design principles by reducing material consumption and weight, thus minimizing environmental impact. In essence, composite configurations represent the cutting edge of material development. Through the use of microstructures and interfaces, composites offer unprecedented capabilities, performance and stability.

Currently, there is extensive research addressing the mechanism of plastic deformation and strong plastic coordination of heterostructure metals. However, there is a relative lack of research on their fracture behavior and toughness mechanisms. This review aims to fill this gap by specifically examining heterogeneous nanostructures and providing an overview of their unique mechanisms for strength and toughness enhancement. It also explores the potential of high-strength components from biological systems known for their exceptional toughness and intricate configurations. This approach holds promise for the development of advanced metal matrix composites that simultaneously improve the strength and toughness of lightweight structures. Since the design of composite architectures plays a crucial role in advanced materials research, the goal is to propose tailored architectures that exhibit exceptional strength and toughness. Therefore, in this review paper, the advantages of toughness, the mechanisms of toughness enhancement, the properties of reinforcement distribution in architectural structures, and the relevant structural parameters are investigated.

## 2. Strengthening and Toughening through Macro-Heterogeneous Configurations of Reinforcement

To explore the concept of heterogeneity, this paper emphasizes the spatial distribution and size variations of reinforcements, considering the representative volume element (RVE) as the macro-scale structural unit [23]. In microstructurally inhomogeneous composites, there are distinct regions: a reinforcement lean region (representing the “soft” phase) and a reinforcement-rich region (representing the “hard/strong” phase) (Figure 1) [24]. It is important to note that different characteristics correspond to the reinforcement-rich or reinforcement-lean phases, rather than the inherent hardness of the ceramic reinforcement itself [24,25]. Achieving both strength and flexibility is a known challenge in material engineering, and typically, metal matrix composites involve a hard phase within a soft matrix. However, certain MMC design configurations, such as TiCp/Ti_6_Al_4_V composites with a network microstructure, have the harder phase as the continuous phase itself [24,26]. The presence of reinforcements not only supports the soft matrix but also hinders crack propagation by acting as obstacles. These composites exhibit heterogeneous microstructures with both soft and hard phases, influencing the final composite’s strength and toughness characteristics at different length scales [27]. Indeed, this aligns with the suggested characterization of heterogeneous materials, which are composed of regions exhibiting significant variations in strength. These regions can vary in size from micrometers to millimeters. To be more exact, heterogeneous materials represent a novel category of substances that exhibit exceptional combinations of strength and ductility beyond the reach of their homogeneous equivalents [28,29,30,31].

In this context, illustrative examples can improve the understanding of this approach. Regarding the geometry of the layers, multi-layer structures exhibit different macroscopic stress and strain behaviors under applied loads [32,33]. These structures effectively control the size of the structural components at the meso- and micro-scales, resulting in significant strength at the interface between the hard and soft phases. In addition, advances in strength and toughness mechanisms influenced by the interface-affected zone (IAZ) [34] or grain-boundary-affected zone (GBAZ) [35,36,37], based on soft phase plasticity gradient theory [29,38], contribute to a deeper understanding of the importance of grain boundary engineering. By modifying and developing structural components and using mechanical models, the optimization of structural designs can be achieved.

The presence of heterogeneity in layered structures plays a crucial role in improving toughness. The orientation effects at the interface between the hard and soft phases, although important, mainly affect the normal direction and may result in lower hardness. To counteract this, some proposed structures increase the width of the soft layer to improve processability and accommodate more dislocations [39,40,41]. Plastic strain is mainly concentrated in the soft coarse grain phase, while the hard ultrafine grain (UFG) phase contributes less to strengthening and toughness [39]. However, such a structure may have limited failure due to premature necking in the soft layer. On the other hand, the multi-core configuration, also known as concrete-like architecture [42], simultaneously provides significant toughness and strengthening effects by reducing the interfacial volume density between the soft and hard phases. Compared to layered composites, the designed composite with a multi-core configuration exhibits higher fracture energy and prevents catastrophic failure at maximum load [43,44]. The gradient configuration, in which the grain size increases from nanoscale at the surface to coarse grains in the core, is a special design that attenuates the stress concentration at the interface, resulting in improved toughness and strength [45,46,47,48]. Incorporating a gradient distribution of reinforcements in the matrix can result in significant improvements in strength and toughness. By varying the concentration or size of reinforcements along specific directions, one can direct the initiation and propagation of cracks away from critical regions. This redirection of crack paths limits catastrophic failure, allowing the material to withstand higher loads and absorb more energy before it breaks. An example of the process of toughening in the scientific literature is the use of the term “toughening,” which refers to the process that makes the material more resistant to cracking [49]. When a crack propagates, the associated irreversible work varies in different classes of materials. Therefore, the best toughening mechanism varies between different materials [49,50]. As mentioned earlier, intrinsic toughening mechanisms are processes that act upstream of the crack tip to increase the toughness of the material. These are usually related to the structure and bonding of the base material as well as microstructural features and additives [51,52]. Another example is the strain distribution around a matrix crack bridged by reinforcing elements and its effect on tensile fracture [53]. The results show that the strain propagating in the elliptical region around the cracks can be controlled by varying the size and height of the reinforcing fibers. This allowed them to redirect the crack paths away from the critical areas, resulting in improved strength and toughness [53]. These are some examples of how the distribution gradient of reinforcement can be added to the matrix to increase strength and toughness. There are many other mechanisms and techniques that can be used to achieve this goal. Although progress has been made, more research is needed to optimize gradient sizes and explore new possibilities in this area.

The primary optimization approach for creating heterogeneous configurations involves distributing the reinforcement within the matrix phase, allowing for the control of the strength of the interface between the soft and hard phases, and optimizing the size based on strain gradient plasticity theory. However, this approach has its limitations as it neglects the interdependence between different reinforcements and structural components such as grain size, reinforcement distribution, shape, and crystal structure of the matrix. If the design plan remains unchanged, then the local stress concentration may still lead to inhomogeneous plastic deformation, resulting in early necking and not effectively overcoming the strength–toughness trade-off. Conversely, reducing the size of the soft phase to a few microns can transform the multi-core configuration into an inverted nacre structure [54], where elongated and corrugated soft constituents are embedded in a matrix of hard constituents. This structure exhibits increased tensile strain and strength, with the strain-hardening ability of the soft phase offsetting the strain softening of the hard phase caused by microcracking. The dispersed soft phase stores plastic deformability until adjacent hard phases form microcracks [44], which are then blunted and prevented from propagating by the complete plastic deformation and hardening of the soft phase at the crack tip [31]. The superior comprehensive strengthening and hardening effect results from the tailored hierarchical structure, which promotes the stable propagation of microcracks by sufficient strengthening and prevents a significant reduction in the final tensile strength.

## 3. Strengthening and Toughening of Micro-Heterogeneous Configurations of Reinforcements

A design configuration consists of microelements that interact to form mesoscopic objects that act as building blocks within the RVE [55]. When we talk about heterogeneous configurations of gain elements, we refer to the relationship between micro- and nano-scale gain elements and their distribution, size, shape, and other structural parameters relative to the matrix material. The terms network, micro–nano, and micro–nano hybrid architectures are used to describe the design scale and configuration. Common to all is the uniform distribution of macro- and micro–nano-scale gains within a given engineered structure. AMMCs illustrate this concept by exhibiting improved grain boundary efficiency compared to base materials or those with crystalline particles. This improvement is attributed to the presence of a three-dimensional network of closed alumina cells at the micro-scale filled with elastoplastic aluminum. This network effectively prevents restoration mechanisms across grain boundaries and grain boundary sliding during hot working, resulting in an extremely stable UFG structure with superior thermal and strain stability [56,57,58]. Such macro-heterogeneous structures hold great potential for the development of materials with desirable properties suitable for load-bearing structural applications, especially at high temperatures.

Previous studies [59,60,61] have shown that the design approach for network configurations has changed from continuous reinforcement networks to quasi-discontinuous and quasi-continuous networks. The original design of the (Al_3_Zr + Al_2_O_3_np)/2024Al continuous network composite exhibited concentrated and agglomerated regions of reinforcement along the grain boundaries of the matrix, resulting in significant stress concentration and limited balance between strength and toughness. Attempts to adjust the sintering temperature, pressure, and other process parameters did not result in a satisfactory balance between strength and toughness [62]. On the other hand, configurations with quasi-continuous or continuous reinforcement networks, such as SiC/Al composites with a 3D quasi-skeleton (interpenetrating network structure), have shown remarkable improvements in thermophysical properties. The geometry of the SiC reinforcement has a significant impact on these properties, and the co-continuous structure of the SiC reinforcement and the Al matrix in 3D SiC/Al composites has been shown to be particularly advantageous [59]. These configurations not only exhibit excellent homogeneity, but also show improved mechanical and elastic properties due to their unique structure and strong interfacial bonding between the matrix phase and the reinforcing phase [60,61].

Compared to conventional composites using particles or fibers as reinforcing materials, composites with an interpenetrating phase network exhibit improved toughness and isotropic microstructures. In MMCs with a quasi-continuous 3D network structure, the interface between the continuous ceramic and metal phases is smaller, resulting in lower interfacial thermal resistance than partially reinforced MMCs with the same ceramic volume fraction [59]. The strategic distribution of reinforcements within a network or quasi-continuous skeleton can simultaneously improve the strength and toughness of titanium matrix composites at room temperature. In addition, the incorporation of a discontinuous three-dimensional graphene network into copper-based composites improves their ability to accommodate geometric dislocations, resulting in improved mechanical properties such as increased elastic modulus, yield strength, tensile strength, elongation at break, and fracture toughness [63]. This type of network configuration provides a uniform distribution of reinforcement throughout the matrix and ultimately improves the strength and toughness properties of the metallic composite. In addition, the optimization of structural parameters such as the size, shape, and distribution of reinforcement is expected to further improve the strength and toughness properties.

The nano-micron hybrid configuration is an effective design that combines two different sizes of reinforcements to synergistically improve both strength and toughness under load. Larger particles can withstand significant loads and accumulate strain energy, triggering recrystallization in the surrounding metal matrix [17,64,65,66]. On the other hand, smaller nanoparticles reduce the stress concentration in the GBAZs and IAZs around the larger particles, retarding phenomena such as recrystallization and grain growth (Figure 2) [3,67,68,69,70]. The uniform distribution of reinforcing elements in two different sizes is challenging in conventional fabrication processes, especially for nanocarbon materials such as carbon nanotubes and graphene, limiting the achievement of desired toughness and plasticity in nano/micron hybrid configurations [71,72,73,74,75].

In order to identify a suitable fabrication technique, step-by-step powder metallurgy, also known as powder assembly, has emerged as one of the most effective and practical methods for achieving the essential structural parameters and harnessing the synergetic effects of reinforcements with different size scales in composites. Employing this approach, the (B_4_Cp+Al_2_O_3_np)/Al micro–nano hybrid composite has been successfully produced, exhibiting an elongation-to-fracture of approximately 8.9%. This value surpasses the elongation-to-fracture of 4% observed in B_4_C/Al composites reinforced solely by micron-scale reinforcements. The presence of sub-micron-sized Al_2_O_3_ nanoparticles within the aluminum grains, forming Orowan loops in the grain interiors, significantly contributes to the hardness and facilitates uniform plastic deformation [43]. Furthermore, the adoption of a reinforcing design configuration leads to enhanced fatigue and strength properties in bimodal-sized Al_2_O_3_/Al nanocomposites, which incorporate both micro-sized and nano-sized Al_2_O_3_ particles. The uniform presence of nano-sized alumina within the grain interior restricts dislocation mobility, while the micro-sized alumina at the grain boundaries limits grain boundary migration [17].

The incorporation of nanoparticles as reinforcements within the matrix grains surrounding micron-sized reinforcements offers a significant enhancement in both toughness and tensile strength. In specific cases, the addition of sub-micron Al_2_O_3_ particles in nano/micron hybrid composites, such as (Al_2_O_3_p+AlN-np)/Al, facilitates the dispersion of AlN nanoparticles. The presence of sub-micron reinforcements activates a portion of the matrix grains, leading to the recrystallization and reorganization of stored dislocations, resulting in simultaneous increases in strength and toughness [58,76,77]. Generally, achieving the uniform dispersion of nanoparticles is a vital concern for making nanoparticle-reinforced composites as it directly impacts the mechanical, thermal, and other functional characteristics of the composite. Here is a list of some common approaches used to ensure homogeneous dispersion of the second phase within the matrix:Mechanical Mixing: Utilizing mechanical methods such as ball milling, attrition milling, and high-energy mixing can help break down agglomerates and promote a more even distribution of the second-phase particles within the matrix.Ultrasonic Dispersion: Ultrasonic waves can be employed to disperse and break down clusters of second-phase particles, ensuring their uniform distribution within the matrix material.Powder Metallurgy (PM) Techniques: Powder metallurgy methods, such as hot isostatic pressing (HIP) and spark plasma sintering (SPS), involve consolidating powders under controlled conditions. These methods can result in a more uniform distribution of the second phase due to the homogenizing effect of powder compaction [64].Liquid Phase Infiltration: Introducing a liquid matrix into a preform of reinforcing particles can help achieve a uniform distribution of the second phase, especially in cases where the reinforcing phase is in a fibrous or porous form.Gradient Precipitation Techniques: Controlling the cooling rate during solidification can promote the uniform precipitation of the second phase, resulting in a more homogeneous distribution.Surface Treatment and Coating: Treating or coating the second-phase particles before incorporation can improve their wetting and dispersion within the matrix material.Electrostatic Dispersion: Employing electrostatic forces can assist in dispersing and separating particles, enhancing their uniform distribution within the matrix.Additive Manufacturing: Techniques such as 3D printing allow precise control over the placement of the second-phase material, enabling the creation of complex geometries with uniform distribution.Melt Infiltration: For fiber-reinforced composites, melt infiltration involves impregnating a porous preform with molten matrix material. This can facilitate a more even distribution of the matrix around the fibers.Controlled Agitation and Stirring: Utilizing controlled agitation or stirring during the mixing and consolidation processes can prevent settling and promote uniform distribution.Sol-Gel Techniques: Sol-gel processes can result in the formation of a homogenous precursor material that can be used to achieve uniform distribution upon consolidation.Smart powder processing: In this method, the process involves utilizing four distinct mechanical devices to induce mechanical forces, leading to the covering of fine shell (guest) particles onto the surface of larger core (host) particles, which in turn develop the advanced production of materials [71,78].

These ideas can be based on the specific type of composite, the composition of the secondary material, and the desired end product. A combination of these methods can also be used to obtain a good view of the second phase in the matrix material. However, achieving the uniform distribution of high-modulus and high-strength nanocarbon reinforcements such as CNTs and graphene in composites remains a challenge for the development of advanced MMCs. The adverse effects, such as lateral reactions and formation of metal carbides at the interfaces, may compromise the structural integrity of CNTs or graphene due to excessive energy input during ball milling during the initial fabrication process. Therefore, minimizing the energy input is proposed as a solution [46]. Moreover, the enrichment of nanocarbon reinforcements significantly affects the plastic behavior of the resulting composite. For example, significant increases in elastic modulus, yield strength, tensile strength, and elongation at break were observed in a (SiCp+CNT)/6061Al nano-micron hybrid composite fabricated by mechanical milling in which CNTs were grown in situ on the micrometer surface of SiC particles [47,48]. The use of a predispersion process, such as high-shear predispersion, has also been shown to be effective in predispersing nanocarbon on the surface of platelet Al powder and providing enhanced protection in subsequent mechanical powder processes [71,79]. A well-coordinated CNT dispersion, structural integrity, and interfacial adhesion in a fabricated 1.5 wt.% CNT/6061Al composite demonstrated improved tensile toughness. These results underscored the importance of structural designs that ensure the reasonable distribution of multi-phase reinforcements while considering the optimal conditions for achieving desired composite properties. The excessive volume fraction of reinforcements in the metal matrix should be avoided, as this may lead to stress concentration at the interfaces and in the GBAZ, resulting in poor mechanical behavior in terms of strength and toughness. In summary, the shape and size of reinforcements play a crucial role in achieving high strength and toughness in composites. It is worth noting that the toughness of composites has an inflection point or reversal at a certain stage.

Inspired by the micro–nano brick structure found in nacre (mother of pearl) [21], efforts have been undertaken to develop a similar configuration in layered composites. This configuration involves the distribution of reinforcements with varying size scales, shapes, and distributions within the matrix. Similar to the brick-and-mortar arrangement, brick-shaped mineral platelets at the micro-scale are embedded in a matrix of nano-scale material. The utilization of flake powder metallurgy technology has played a crucial role in achieving precise control over the configuration parameters in layered composites [73,80]. By producing platelets in two dimensions, this approach ensures better geometric compatibility, leading to a more uniform distribution of reinforcements. Furthermore, the elongated grains with an increased aspect ratio (width in the nanometer range and length in the micrometer range) enable the storage of dislocations. This facilitates the intentional placement of in situ Al_2_O_3_ nanoparticles, CNTs, graphene plates, and other reinforcements at the grain boundaries of the elongated grains. As a result, a high density of geometrically necessary dislocations (GNDs) is formed near the grain boundaries and interfacial zones, compensating for the loss of strain hardening caused by ultra-fine grains [81,82].

Various techniques have been employed to enhance the mechanical entanglement and load transfer effect at interfaces in composite materials. These methods include combining powder metallurgy (PM) with conventional rolling [3,72], electrophoretic deposition of CNTs on metal foil [83], constructing biomimetic mineral bridges, and in situ reinforcement growth [84], among others. By improving the interfacial bonding and structural quality of the reinforcement, these techniques require more energy for crack propagation, resulting in a significant increase in toughness.

The geometric compatibility between two-dimensional graphene plates and flake-shaped building blocks formed in the flake PM process makes graphene an ideal reinforcement for biomimetic brick matrix structures. For example, in the GNS/Al micro–nano brick composite, the geometric compatibility of graphene plates with flake-shaped platelets prevents the annihilation of dislocations at the grain boundaries [85,86]. This composite demonstrates an increase in tensile strength from 201 MPa to 302 MPa, while maintaining a similar level of elongation to fracture.

Recently, a new design approach for high-performance Ti_2_AlC/TiAl composites with a micro–nano-laminated configuration has been explored. This design exhibits a favorable combination of compressive strength (1807 MPa) and ductility (25.5%) at room temperature [87]. The composite consists of microlaminated Ti_2_AlC, micro–nano-γ-TiAl/α2-Ti3Al lamellar colonies, and nano-laminated Ti_2_AlC. These structural features contribute to the increased strength of the Ti_2_AlC/TiAl composite through load transfer reinforcement and Orowan strengthening effects commonly observed in metal matrix composites. In addition, this design configuration improves the ductility and toughness of the Ti_2_AlC/TiAl composite through several plastic deformation mechanisms [87]. The configuration of a design, including the size and shape of its components such as grains and particles that provide reinforcement, plays a critical role in the ability to store dislocations and the likelihood of defects forming in the structure. The response to high strain rates is influenced by the constraints imposed by the geometry of the reinforcement [48,73,88]. In addition, the interaction between intrinsic mechanisms (related to plasticity) and extrinsic mechanisms (related to shielding) to increase toughness is strongly influenced by the design configuration [89,90,91]. Intrinsic mechanisms act primarily on small scales (in the nanometer or micrometer range) upstream of the crack tip and enhance the material’s inherent resistance to damage, thus increasing both crack initiation and propagation toughness. In contrast, extrinsic mechanisms produce a shielding effect at larger length scales, usually behind the crack tip, and only affect toughness during crack propagation [92]. Essentially, intrinsic toughness relies on plasticity to increase a material’s resistance to damage, resulting in improved crack initiation and propagation toughness. On the other hand, extrinsic toughness reduces local stresses and strains at the crack tip and is effective only in the presence of a crack, affecting crack propagation toughness. In this context, the combination of multi-scale extrinsic toughening mechanisms [35,83,93] induced by the reinforcements and the matrix can significantly contribute to increased strain hardening and the formation of exceptional strength and toughness.

In configurations featuring micro-scale heterogeneity of reinforcements, incorporating nanosized reinforcements alongside sub-micron and micron reinforcements leads to improvements in strength and toughness across various size scales, ultimately enhancing the balance between stability and toughness. The utilization of nanoscale reinforcements in future advanced composites is considered crucial, as indicated by numerous studies [94,95]. For instance, a boron carbide (sn-B_4_C)/Al composite with spatial arrays of sn-B_4_C demonstrated a significant increase of 26% in tensile strength and 30% in toughness. This enhancement can be attributed to the presence of fiber-like nanoparticle-rich (NPR) zones, acting as “hard” fiber-like units, effectively sustaining tensile loading and enhancing the strengthening efficiency of sn-B_4_C. Furthermore, the arrangement of NPR zones, surrounded by nanoparticle-free (NPF) zones as the softer phase, contributes to improved strength with minimal loss in ductility [94,95,96].

The reinforced design configuration, particularly the fiber-like distribution of reinforcements, introduces additional microstructural strengthening through dislocation strengthening and the activation of Orowan mechanisms. The presence of spherical nanoparticles within the NPR zones immobilizes more dislocations, leading to a significant disparity in hardness values between the NPR and NPF zones. This disparity enables stress transfer from the “soft” NPF zones to the “hard” NPR zones, resulting in the extraction of some fiber-like NPR zones at the fracture surface during tensile loading. However, challenges persist in achieving uniform distribution and limited work hardening in ultra-fine grains. Controlling the spatial distribution of reinforcing particles at the micro- and meso-scale to engineer distinctive microstructural configurations has emerged as an innovative approach to overcome existing trade-offs in material properties [36,97,98].

Considering the toughness of micro-heterogeneous reinforcement configurations, it is clear that refining the interaction between the matrix and the reinforcement phase can lead to remarkable improvements in the toughness of CNT- or graphene-reinforced MMCs. Yizhang Liu et al. [99] discussed the development stage of ceramic matrix composites reinforced with nanocarbon materials. The study investigated the influence of interfacial bonding conditions on the fundamental properties of these composites and explored strategies to improve this interfacial bonding. Afifah Md Ali et al. [100] addressed three key topics on the challenges associated with the use of graphene as a reinforcing material. These topics included the mechanisms by which graphene increases the strength of metal matrices and the constraints that limit the extent of property enhancement. Surface modification techniques applied to the reinforcing phase can optimize interfacial adhesion. Functionalization of the surfaces of CNTs or graphene with suitable chemical groups can improve compatibility with the matrix, ensuring robust load transfer and preventing premature detachment. Rama Dubey et al. [101] investigated how modifying the surfaces of CNTs can optimize their interaction with other materials. Introducing specific chemical groups into CNT surfaces increased their compatibility with surrounding matrices, resulting in robust load transfer and prevention of premature detachment. In addition, the development of hybrid structures in which CNTs or graphene are coupled with secondary reinforcements such as nanoparticles or fibers can result in multiple toughening mechanisms. This synergistic combination of reinforcing elements leads to mechanisms such as crack deflection by secondary phases, crack fixation by nanoparticles, and stress redistribution by fiber bridging. The resulting interactions mitigate crack propagation and contribute to the overall improvement in toughness. Jyoti et al. [102] investigated the field of hybrid structures where graphene or carbon nanotubes were combined with additional reinforcements such as nanoparticles or fibers. This integration resulted in various toughening mechanisms, including the deflection of cracks by secondary phases, immobilization of cracks by nanoparticles, and redistribution of stresses by fiber bridging. The combined effect of these mechanisms impedes crack propagation and increases the overall toughness of composites.

It is important to note that both macro- and micro-scale configurations of reinforcing elements significantly affect the balance between toughness and strength in metal composites. However, achieving a satisfactory spatial distribution of reinforcements requires careful consideration. For example, achieving uniform intragranular dispersion of CNTs in an aluminum matrix requires specific conditions related to time, energy, external loading, and thermal activation [103,104]. In addition, challenges remain in overcoming the trade-off between strength and toughness, highlighting the need for further extensive research in the area of strengthening and toughening mechanisms. In summary, the relative decrease in toughness observed in CNT- or graphene-reinforced MMCs requires a dual strategy involving both macro- and micro-heterogeneous configurations of reinforcements. Strategic design choices, such as gradient distributions, sacrificial phases, surface functionalization, and hybrid structures, can leverage toughness enhancement mechanisms to significantly improve the overall performance of these advanced composites.

## 4. Strengthening and Toughening the Matrix through Intrinsic Mechanisms

The enhancement of toughness and strength in metal matrix composites depends not only on the uneven distribution of multiple reinforcing phases at different length scales but also on the structural parameters of the design configuration. Heterogeneous structures consist of diverse microstructural components combined at interfaces with strong bonding. The incorporation of alternating strength and toughness in the design configuration of heterogeneous metal matrix composites effectively improves their toughness, and the specific toughening mechanisms are influenced by factors such as crack orientation, interface strength, and microstructural characteristics of the components. These structures exhibit notable variations in mechanical properties on either side of the interface due to the presence of multiple heterostructures. Typically, the greater the difference in mechanical properties, such as hardness and strength, between the components, the more significant the strengthening effect of layered materials. The strengthening effect in heterostructured metals is attributed to the incompatibility of deformation between the constituent units, leading to a prominent strain gradient near the soft/hard interface. However, certain structural features, such as the ultra-fine grain size of the matrix, can restrict the ability to undergo work hardening, consequently affecting the improvement of strength, toughness, and plasticity in the resulting configuration.

In conventional MMCs with grain sizes ranging from a few micrometers to several hundred micrometers, both plasticity and toughness usually increase simultaneously. However, in specially designed composites consisting of ultra-fine grains, a trade-off occurs in which toughness and plasticity decrease while strength increases. This trade-off is attributed to the reinforcing contribution of grain boundaries and interfaces achieved by effective grain refinement processes during powder metallurgical fabrication [105], zener pinning effects [68,106,107,108], and other factors that result in grain refinement to ultrafine or even nanoscale sizes. The creation of conditions for plastic deformation, such as twinning or the introduction of defects in the atomic arrangement at low temperatures and high stresses, is comparatively difficult and almost impossible compared to other deformation techniques [105]. The presence of coherent low energy twin boundaries effectively prevents dislocation motion, strengthening MMCs while maintaining an acceptable level of plasticity and strain hardening [109]. This advantage applies not only to uniformly structured nano-twin metals [110,111], but also to the strengthening and toughening of heterogeneous structures in MMCs [112].

It is important to recognise that stress-induced transformation by plastic deformation is only possible in certain composites with specific compositional and microstructural conditions [113]. Various mechanisms, such as grain rotation, grain boundary migration, grain boundary slip, and dislocation ascent, affect the ability to strain harden at both room temperature and high temperatures and thus play an important role in determining strength and toughness properties [114,115,116]. However, the presence of these mechanisms often weakens the ability for intragranular strain hardening, resulting in lower toughness and plasticity. Therefore, to achieve uniform strain with minimal loss of strength properties, the decrease in strain hardening must be compensated for in ultra-fine or nanoscale grains.-

Recent studies [81,82,83] have shown that two critical factors, namely inherent heterogeneous deformation induced (HDI) strain hardening and HDI stress, have no tabular effect on improving the strength–toughness ratio in nanoparticle reinforced MMCs. According to this theory, grains with multiple scales exhibit different plastic deformability, which enables HDI activation. This leads to strain partitioning, which results in back stress hardening in the softer zones and forward stresses in the harder zones, contributing to HDI hardening. In other words, during plastic deformation of heterostructured materials, the heterogeneous zones deform non-uniformly, which is probably responsible for the hardening and additional strain hardening observed in heterogeneous materials [117,118]. This mechanism allows an optimal combination of strength and toughness properties.

HDI hardening arises from the interaction between the hard and soft zones, creating an elastic-plastic situation at the micro-scale during deformation. Within the soft domain, GNDs are impeded and accumulate at domain boundaries, leading to the development of long-range internal stress known as back stress. This back stress is believed to contribute to the enhanced strength of heterostructured materials [28]. On the other hand, in the hard zones, stress concentrations occur at the zone boundaries due to the accumulation of GNDs, generating forward stress with magnitudes several times higher than the applied stress [118]. The accumulation of dislocations in the soft phase induces forward stress in the hard phase, establishing a balance in mechanical properties. The specific stress states that emerge at the interface between the hard and soft zones, influenced by the mechanical behavior of the hard phase relative to the soft phase, significantly impact the material’s toughness and strength. The back stress and forward stress are coupled at the domain (or grain) boundary and act in opposing directions. Importantly, the forward stress is induced by the back stress, and it can be logically deduced that the exceptional combination of strength and ductility observed in heterogeneous MMCs with a well-designed configuration is attributed to the back stress, aligning with the assumptions made in [117,118,119,120].

The ongoing research in the field of heterogeneous MMCs focuses on investigating internal factors associated with design configurations and their impact on material properties [121]. The intrinsic design of the matrix in heterogeneous MMCs is influenced by factors such as grain shape, size, and interface bonding, which play significant roles in work hardening, crack initiation and propagation, as well as the resulting strength and toughness behavior. The inclusion of nano-sized reinforcements at grain boundaries, known as the Zener pinning effect, offers effective control over grain size compared to other factors such as alloy solutes, dislocations, and precipitates [107,108,122]. As a result, it is anticipated that MMCs can provide greater freedom in structural design and mechanical property control.

Early attempts at bimodal designs in metal composites relied on a trial-and-error approach, lacking a comprehensive understanding of the mechanisms behind strengthening and toughness, leading to misconceptions about fracture mechanics in such configurations. However, by optimizing the input energy in processes such as ball milling [58,123], friction stir processing [68,124], or incorporating reinforcing nanoparticles, it becomes possible to regulate the grain size of the matrix and create a bimodal grain structure, thereby enhancing the toughness and strength properties of the composite. A proper allocation of tasks between reinforcement particles and the matrix can effectively generate bimodal-sized metal matrix composites by introducing bimodal-sized SiC particles [125]. The presence of multi-scale reinforcement significantly influences the development of the bimodal structure, where regions without coarse SiC particles represent the coarse-grain regions, and grain sizes around nano SiC particle bands represent the fine-grain regions. This approach achieves a favorable balance between strength and toughness, benefiting from grain refinement, Orowan strengthening, coefficient of thermal expansion (CTE) strengthening, uniform distribution of nano SiC particles, and the presence of coarse grains [40,125]. While the achieved tensile plasticity may not be exceptionally high [69,125], it has been reported that the back stress strengthening effect in the elongated coarse-grained region surpasses that in the spherical coarse-grained region [126]. However, further investigation is still required to quantitatively optimize the relationship between the coarse-grained region and the sub-micron-sized microcrystalline region [127,128].

Currently, the focus of grain size optimization in the design concept revolves around determining the size of the IAZ, which refers to the zone with a strain gradient [34]. This determination is based on the theory of strain gradient plasticity [26] and the intrinsic engineering toughness derived from linear elastic fracture mechanics. In heterogeneous MMCs, the characteristic IAZ is formed near interfaces due to the density gradient of GNDs. When plastic deformation occurs, dislocations emitted from sources at the interface create the IAZ, which typically has a length scale on the order of several micrometers. While the width of the IAZ remains relatively constant with increasing tensile strain, the strain intensity within the IAZ increases, resulting in a higher strain gradient and work hardening through back-stress effects [34]. The optimal spacing of the IAZ plays a crucial role in achieving a high level of intrinsic strength–toughness, as illustrated in Figure 3 and demonstrated in studies on composites such as CNT/Al [39,72,129]. It has been found that in a heterogeneous configuration of matrix grains, the best combination of toughness and tensile strength is attained when the width of the coarse grain region is reduced to approximately twice the width of the IAZ [60]. Another approach is to have the diameter of the spherical coarse grain region equal to the plastic zone at the tip of the nearest crack [61,62]. In a bimodal grain structure with numerous interfaces between UFG and coarse-grained regions, a high density of GNDs is accumulated at the heterogeneous interfaces during deformation due to their incompatibility in deformation [34]. This accumulation of GNDs gives rise to the formation of the IAZ, effectively enhancing work hardening through multiple dislocation-mediated mechanisms. The volume percentage of these interfaces significantly influences the work hardening induced by heterogeneous deformation at the soft/hard phase interface areas.

Moreover, the combination of flexibility, toughness, and high strength can be attained through a trimodal grain configuration in powder metallurgy produced MMCs. This configuration involves the distribution of fine grains between coarse and ultra-fine grains, resulting in a trimodal grain structure. Such a configuration helps reduce stress concentration and inhibit strain localization within the microstructure [39,130]. For instance, in CNT/2024Al composites, the trimodal grain configuration exhibited significantly higher yield strength (561 MPa), tensile strength (723 MPa), and uniform elongation (6.7%) compared to the bimodal grain configuration with a yield strength of 532 MPa, tensile strength of 625 MPa, and uniform elongation of 3.8% [131]. The presence of multi-scale features in the trimodal grain structures effectively contributes to the exceptional strength–toughness balance observed in trimodal MMCs.

It is worth noting that microstructures with gradient designs hold promise for stability and toughness enhancement [85]. However, achieving optimal mechanical properties through gradient configurations in MMCs remains challenging. Conventional processing techniques struggle to precisely control structural gradients across a range from nanoscale to macroscale, leading to limitations in achievable gradients. Unlike alloys, achieving precise grain gradients similar to those in alloys is difficult due to technical issues in preparing gradient configurations and the pinning effect of reinforcements on grain boundaries. Deformation at the scale of structural gradients differs fundamentally from deformation in conventional metallic materials [85]. This type of deformation exhibits unique dislocation behavior, interface-related phenomena, and interactions between GNDs and interfaces. Furthermore, the inhomogeneous deformation of gradient nanostructures results in the development of a long-range HDI stress field [28,132].

**Figure 3 materials-16-05745-f003:**
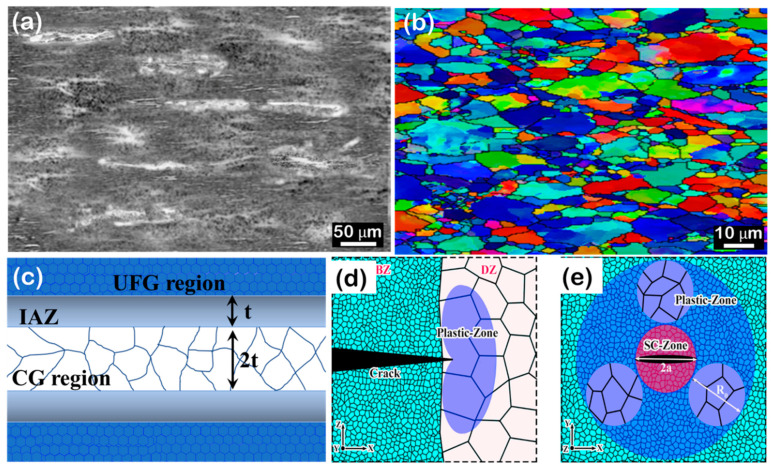
(**a**) OM images of heterogeneous configuration of Al–Mg/1.5 wt.% CNTs in the as extruded condition; reprinted from [39], with permission from Springer (**b**) IPF map showing the grain structure of Al–Mg/1.5 wt.% CNTs; reprinted from [39], with permission from Springer (**c**) Schematic of interface affected zone (IAZ) in the trimodal configuration in Al–Mg/CNTs composites, (**d**,**e**) Schematic of toughening mechanism in heterogeneous materials; reprinted from [133], with permission from Elsevier.

Currently, the primary focus of research lies in controlling structural parameters, although their influence on mechanical properties remains unclear. For example, while the presence of “soft” layers is believed to enhance plastic deformation capability, an excessive number of such layers can compromise the strength of the composites [134,135]. In the case of a bimodal design configuration, narrow coarse grain bands offer greater advantages in terms of toughening and strengthening compared to wider or ultra-narrow coarse grain bands, mainly due to their effective microcrack blunting effect [136]. In heterogeneous MMCs design configurations, cracks tend to initiate in brittle zones (BZs) rich in reinforcements due to their limited ductility. However, as these microcracks propagate into ductile zones (DZs) free of reinforcements, plastic deformation occurs in the DZs, effectively blunting the cracks [133]. This elastic-plastic deformation behavior near the microcrack tip, as per linear-elastic fracture mechanics, aids in blunting the microcrack, preventing further propagation, and achieving a favorable balance between strength and toughness. Furthermore, the strategies for adapting structural parameters and optimizing the interaction between structural components, such as reinforcements, matrix grains, and affected regions, are not yet well-documented and require comprehensive further research.

It is important to note that the design resulting from the combination of two-scale reinforcement and a bimodal-sized matrix structure is not restricted to dual matrix micro/nano hybrid structures (Figure 4a,b). Theoretically, any reinforcement configuration can be related to the matrix configuration (Figure 4c,d). In this context, dual gradient configurations have been developed within the framework of composite metal matrices [8], warranting further investigation. The gradient distribution of reinforcement particles within the metallic matrix enhances the material’s mechanical properties (Figure 4e) and effectively hinders the localized rapid growth of surface nanocrystals. This provides an effective approach for inhibiting grain growth during the preparation of advanced MMCs using powder metallurgy techniques [137].

Additionally, in the investigation conducted by Wu et al. [138], a functionally graded AA7075-B_4_C composite featuring a multi-level gradient structure that encompasses reinforcements, dislocations, and precipitates was examined. The outcomes revealed exceptional bending strength, approximately 1291 MPa, under tensile loading of AA7075, attributed to robust interlayer bonding and the presence of multi-level gradient structures. Furthermore, this multi-level gradient B_4_C/Al composite showcased high strength without significant interlayer cracks or microstructure deterioration post-deformation, contributing significantly to its overall toughness. Li et al. [139] reported the strain hardening of gradient IF steels by incorporating GNDs that are associated with the non-uniform deformation as experimentally observed on the lateral surface. Their results demonstrated that the strength-ductility performance of gradient samples is remarkably enhanced as compared with their homogeneous counterparts. Furthermore, titanium matrix composites reinforced with continuous carbon fibers showed the correlation between gradient configurations in fiber distribution and the resulting improvements in mechanical properties [140]. In general, in order to enhance the design concept for optimizing composite configurations and determining a suitable process path, it is crucial to identify and specify the strengthening and toughness mechanisms involved in developing configuration designs, particularly those primarily achieved through powder metallurgy techniques.

## 5. Designing and Fabricating Composite Configurations Based on Energy Dissipation

The ability of a design configuration to undergo limited deformation is crucial in enhancing fracture toughness as it allows for the localized dissipation of highly stored energy that would otherwise reduce the material’s damage tolerance [141]. The effectiveness of energy dissipation strongly depends on the specific design configurations used [142]. Although synthetic composite design configurations, particularly in metal-based composites, have made some progress in addressing toughness and strength challenges, most advancements have been achieved through experimental research and trial-and-error methods [143]. Energy dissipation during the creation of composite configurations is closely related to the material properties, mechanical forces, and intrinsic properties of the composite materials. Advanced modeling techniques are essential to measure the removal of this force.

Numerical simulations, such as finite element analysis (FEA) and molecular dynamics (MD), provide a virtual platform for predicting the behavior of designed configurations of MMCs under load conditions. These simulations provide insights into how energy is absorbed and localized within the composite, contributing to enhanced fracture toughness [144]. Furthermore, experimental validation helps quantify the energy dissipation. Techniques such as mechanical testing, dynamic impact testing, and acoustic emission analysis provide empirical evidence of energy dissipation during loading. By combining experimental results with theoretical predictions, it enriches the understanding of energy dissipation mechanisms is enriched, allowing for the refinement of design configurations to optimize toughness [145,146]. The new research focuses on uncovering the scientific principles that govern energy dissipation in composite configurations. On the other words, the “how and why” of energy dissipation mechanisms leads to the development of general theories to guide design choices.

The combination of theoretical insight, numerical simulation, and experimental validation empowers researchers to quantify and optimize energy dissipation mechanisms within composite configurations [147]. More importantly, the influence of microstructural components such as interfaces, dislocations, cracks, and local plastic deformation fields on energy dissipation cannot be underestimated. It is important to fine-tune the microstructures at different scales. Advances in nanotechnology and materials science have made it possible to combine macroscopic and nanoscale levels to enable optimization of energy dissipation mechanisms [148,149]. However, it is recognized that the development of multi-stage processes and multiple models presents challenges with “shape control” and “controllability.” Achieving controlled fabrication necessitates a deep understanding of materials processing techniques, innovative manufacturing methodologies, and collaborative interdisciplinary efforts [150].

Existing mechanical theories are not sufficient to fully explain the complex relationships between structural components, forces, and various factors that occur in different configurations. Further investigation of the relationships between properties and components in such configured composites reveals problems with phase criteria and uncertain toughness mechanisms. It is important to note that simply mimicking natural biomaterials and recognizing their exceptional properties will not lead to the same level of performance without a detailed understanding of the underlying reasons and mechanisms. Therefore, there is an urgent need for innovative research to uncover the fundamental scientific principles that simultaneously improve strength and toughness. This research should focus on developing theories and technologies for new configurational designs in advanced metal matrix composites that provide comprehensive answers to the “how and why” behind the emergence of superior properties, particularly high strength and toughness.

The principles of fracture mechanics state that the mechanical behavior of materials involves the dissipation of energy acting externally on their structures. Understanding this concept has shown that engineering design configurations can significantly improve the mechanical behavior and toughness of metallic materials and metal matrix composites by utilizing various energy dissipation mechanisms [151,152]. Tailoring design configurations in heterogeneous metal matrix composites has a profound effect on crack propagation and deformation, leading to effective energy dissipation and increasing the toughness of the composite [153,154]. Inspiration from bio-logical systems, known for their exceptional strength and toughness, can be a fundamental approach to optimize composite configuration plans to achieve maximum strength and toughness [155]. With respect to energy dissipation, the development of energy dissipation theory should focus on two aspects to improve the optimization of composite configuration design strategy.

The first aspect is to prevent cracking and maximize dissipation of static plastic deformation energy. This is achieved by designing the composite surface and overall configuration to allow multiple energy dissipation mechanisms with overlapping deformation processes. By allowing all components of the composite to participate in the energy dissipation process, localized concentration of deformation is effectively avoided and the formation of cracks is prevented. By shaping the structural parameters of the design configuration, energy dissipation is particularly promoted at preferential sites for void formation, reducing the ability to form voids [51]. The second aspect focuses on limiting crack propagation to maximize energy dissipation at the crack tip. The design of composite configurations occurs at the micro- and nano-levels and includes heterostructures such as non-uniform interfaces and multi-scale reinforcements. These factors have a significant effect on the crack tip and its propagation path, activating toughening mechanisms such as crack tip shielding, crack redirection, and bridging. In contrast, crack propagation in a configured design in MMCs involves an additional energy input that impedes or even prevents extension. However, when fabricating configured composites with the goal of maximizing energy dissipation, it is critical to fully consider the mechanism of action of the microstructure, including the effects of defects (such as interfaces, dislocations, cracks, localized plastic deformation zones, etc.) within the material system on deformation behavior and energy dissipation. This requires precise regulation of the fine structure of composite configurations at both scale and local levels to ensure that the composite configuration meets mechanical performance and energy consumption requirements. However, due to the complexity of the composite system, the controlled fabrication of multi-phase and multi-scale configurations faces technical challenges primarily related to “shape control” and “controllability”.

The first challenge relates to differences in size, density, morphology, and surface/interface properties between the multi-phase and multi-scale reinforcement phases and the matrix. These discrepancies lead to inconsistent dispersion conditions and make it difficult to achieve effective dispersion of the components using the same preparation process. Consequently, the production of controlled composite configurations is hindered. Current technologies rely on applying sufficient energy to the composite system to promote uniform dispersion. Methods such as ultra-sonic stirring, high-energy ball milling, friction stirring [156,157,158,159], or laser remelting [160,161] are used. However, it is important to note that additive manufacturing techniques used to fabricate design configurations are limited in their wide application due to factors such as high cost, limited applicability in fabricating large structures and for mass production, inferior and anisotropic mechanical properties, and limitations in starting materials [161,162]. Although these top-down dispersion and composite methods can achieve satisfactory dispersion effects, they fail to produce an ordered bond between micro- and nanoscale reinforcement phases, and the matrix often becomes metastable. As a result, sub-sequent heat treatment can lead to structural disorder and transformation.

In fact, as a critical process in materials engineering, heat treatment has a significant impact on the microstructure and thus on the performance of these composites. It can lead to profound changes in properties such as strength, ductility, hardness and even fracture toughness. In the context of this review, it should be mentioned that heat treatment is an important tool to influence the mechanical behavior of heterogeneous MMCs. By subjecting these materials to certain thermal cycles, it is possible to modify grain size, phase distribution, dislocation density, and even the presence of residual stresses. These modifications often lead to improved properties, such as a better strength/ductility ratio or higher fatigue resistance. For instance, in the work of Bingzhu et al. [163], the heat treatment of a titanium-based composite led to a refined microstructure and a subsequent increase in tensile strength due to the dissolution and precipitation of certain phases. Similarly, the investigations by Myriounis and co-authors [164] on AMCs revealed that heat treatment played a pivotal role in optimizing the interfacial bonding between the matrix and reinforcement, thereby enhancing overall mechanical properties. Hence, it is crucial to develop new principles and approaches for preparation technologies that address the challenges associated with synergistic dispersion of multi-phase and multi-scale reinforcements and the synergistic regulation of the matrix structure. The ultimate goal is to achieve “shape control” in configurational composite preparation.

The second obstacle concerns the composite interface, in particular the interface between the matrix and the reinforcing phase, which is essential for achieving increased strength and toughness. The regions affected by the composite interface, such as the interfacial reaction affected zone (IAZ) and the grain boundary affected zone (GBAZ), exhibit higher dislocation density and thus contribute significantly to energy dissipation [165]. In addition to controlling the structure and bonding state of the composite interface, an effective approach is to activate dynamic energy dissipation mechanisms, including interfacial migration and interfacial reconstruction. It is desirable to have an interfacial phase with a high energy consumption mechanism that can undergo phase transformation [166] or special reactions [167]. This introduces energy dissipation mechanisms such as deformation-induced phase transformation and reaction self-healing.

The basic concept for the preparation of multi-phase and multi-scale composite configurations should follow the principle of “primitive assembly—limited composition—hierarchical structure” [95]. This means that composite structural units with different properties are assembled, smaller units are scaled, and configurational metal matrix composites with different structural units are hierarchically built. By integrating high-throughput characterization technology with multi-scale simulation [168,169], it is expected that precise and localized control of composite configurations can be achieved. This advance will enable the design and fabrication of new patterned metal matrix composites with exceptional performance based on energy dissipation theory.

The effective design of architectural configurations is a fundamental paradigm in the field of MMC and offers a transformative solution to one of the greatest challenges in materials science. The interplay between strength and ductility has long been a difficult trade-off, limiting the optimization of mechanical properties. Design configurations have proven key to overcoming this limitation. By strategically adjusting the arrangement of each phase, MMCs can achieve a harmonious balance between increased strength and improved ductility. These configurations redirect crack paths, promote energy dissipation mechanisms, and ingeniously manipulate microstructural components, all culminating in improved fracture toughness and mechanical integrity. The examples presented in Table 1 emphasis the central role of design in reshaping material behavior and prove that architecture can indeed circumvent the historical trade-off between strength and ductility. This avenue of exploration not only advances the field of MMCs, but also enriches the broader landscape of materials engineering by offering a new dimension in the search for multifunctional, high-performance materials for diverse application requirements.

## 6. Summary and Outlook

### 6.1. Summary

In summary, this review article explores the structural design of MMCs, with particular emphasis on the design configurations of MMCs in improving strength and toughness. The main structural parameters such as shape, size, spatial distribution as well as variety of non-uniform structures are important driving factors when analyzing the evolution of the configuration of MMCs. This explanation leads to the future of theory: exploring energy distribution and deformation from a new perspective beyond the localization constraints. At the same time, the field of quantitative analysis gained importance, providing a complete picture of the strength and toughness mechanisms within composite materials. Expressing the interaction of energy between scales is meaningful, showing energy dissipation between scales as a determinant of crack initiation and propagation. This understanding lights the way for changes in structural dynamics, material strength, and behavior while orchestrating inverse design trajectories for multi-phase, multi-scale reinforced phase, and matrix composite configurations/interfaces.

With development continuing, the principle of combining technology based on the configuration of various components takes the lead. This principle, which led to the careful design of composite configurations/interfaces at micro- and macro-scales, must overcome the well-established strength–toughness reversal dilemma in MMCs. The peak is achieved through a breakthrough in the distribution of stress and strain at the same level, which has the potential to overcome the obstacle that prevents the visualization of energy and synergy. This paper discusses the potential for deformation in MMCs by examining the interaction of settings from a basic to an advanced level, through the perspective of innovation.

In composites, the integration of different matrix materials with different reinforcement materials exhibits anisotropic behavior that leads to responses under different loading conditions. The inconsistency in the provided information creates challenges when estimating and improving mixed models. Addressing these challenges requires a comprehensive approach that takes various factors into account. Effective modeling methods that consider variations in behavior play a crucial role in simulating interactions within composite materials. Additionally, employing optimization algorithms assists in determining the optimal setup for specific load scenarios, enhancing the performance of materials. Practical tests are necessary to bridge the gap between theoretical concepts and real-world outcomes. The validation process serves to confirm not only the accuracy of the model’s predictions but also reveals subtle interactions between features that might not be predicted through theory alone. However, the effort to resolve these issues extends beyond specific fields. Collaborative efforts across disciplines, such as information science, mechanics, and engineering, result in novel solutions. By combining diverse skills and perspectives, scientists and designers collectively gain fresh insights to unravel the intricacies of composites.

As the market for interconnected devices continues to grow, the work of scientists and engineers remains focused on innovation. This ongoing process aims to not only address challenges stemming from variations in behavior and diverse properties but also to formulate a dependable mixture that fulfills the requirements of various businesses.

### 6.2. Outlook

Future research should focus on the development of quantitative analytical methods for energy dissipation and non-localization of deformation to provide a comprehensive understanding of strength and toughness mechanisms in composite materials. The use of energetic criteria for mechanical properties and service behavior can guide the reverse design of composite configurations and interfaces. This will enable the realization of multi-scale composite technologies and overcome the bottleneck in the inversion of strength and toughness in MMCs. Researchers should continue to explore strategies to tailor strength and toughness to address the inverse relationship between these properties in MMCs. In addition, insights from the architecture of biological composites, such as bone, can provide valuable suggestions for achieving exceptional strength and toughness in synthetic composites through hierarchical organization and the combination of different building blocks. Understanding fracture behavior and toughening mechanisms in heterogeneous nanostructures will be instrumental in advancing the field of MMCs. Overall, composite architecture design holds immense potential for tailoring MMCs with exceptional strength and toughness, meeting the demands of lightweight engineering structures across various industries.

## Figures and Tables

**Figure 1 materials-16-05745-f001:**
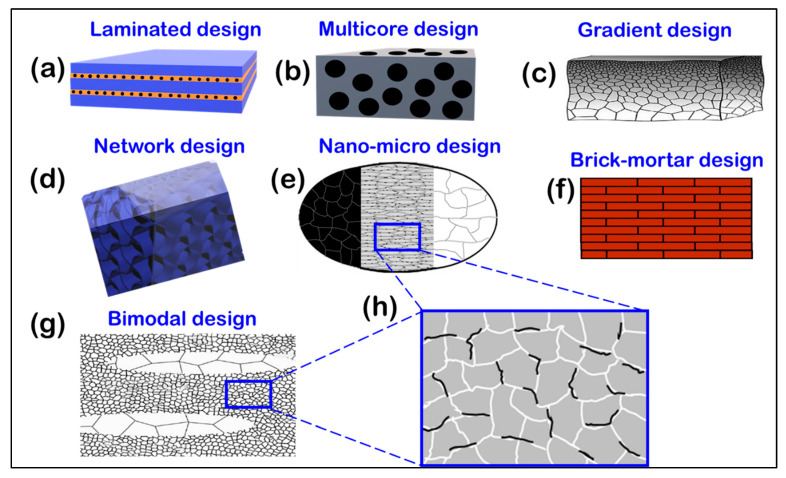
Illustration of the typical heterogeneous design configurations in metal matrix composites, (**a**)Laminated design, (**b**) Multicore design, (**c**) Gradient design, (**d**) Network design, (**e**) Nano-micro design, (**f**) Brick-mortar design (**g**,**h**) Bimodal design.

**Figure 2 materials-16-05745-f002:**
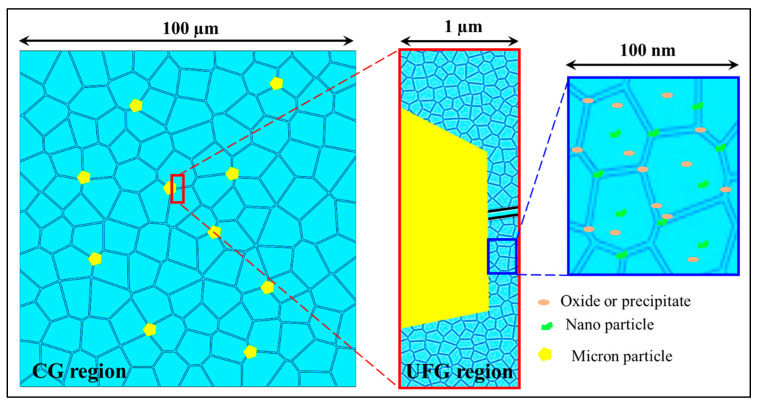
Schematic of the micro–nano hybrid architecture; size-dependent design and cooperative dispersion of micron particles and nanoparticles in the micro–nano hybrid configuration.

**Figure 4 materials-16-05745-f004:**
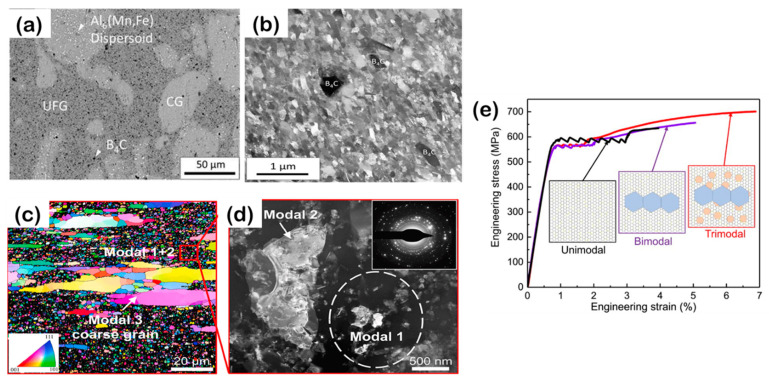
(**a**) Enlarged view of the microstructure showing UFG matrix with B_4_C (dark particles) and CG regions with Al_6_ (Mn,Fe) dispersoids (bright particles); reprinted from [134], with permission from Springer; (**b**) STEM annular dark field image of the UFG matrix; reprinted from [134], with permission from Springer (**c**) IPF map showing trimodal grain structures; reprinted from [126], with permission from Elsevier; (**d**) TEM dark-field image of modal 1 and 2 grain structures, insert is the corresponding SAED pattern; reprinted from [126], with permission from Elsevier; (**e**) Representative engineering stress–strain curves of CNT/Al–Cu–Mg composites with different grain structures, [131].

**Table 1 materials-16-05745-t001:** Designed metallic matrix composites and their mechanical properties.

Architecture Design	Component		Ultimate Tensile Strength/Yield Strength (MPa)	Elongation (%)	Reference
	Matrix	Reinforcement/Content			
Bioinspired Nanolaminated Structure	Al	Reduced Graphene Oxide (RGO)/1.5% vol%	302/263	5.3	[85]
Heterogeneous architecture containing bimodal-grained Al matrix	Al	Micron-B_4_C + CNT/15 wt.% + 1.5 wt.%	475/312	8.8	[91]
Elongated ultra-fine grained (UFG) structure	Al	Al_4_C_3_ + Al_4_O_4_C/0.6 wt.% + 5.6 wt.%	344/282	15.7	[170]
Bimodal-size grained microstructural containing bimodal sized SiC particles	Magnesium	Nano SiC + micron SiC/1 vol% + 9 vol%	402/323	8.3	[125]
UFG structure containing spherical nanoparticles of B_4_C	5083 Al	Nano-B_4_C/2.5 vol%	-/761	2%	[44]
Bimodal structure containing alternative fiber-like nanoparticle-rich (NPR) zones and nanoparticle-free (NPF) zones	Magnesium	Nano-TiB_2_/20 wt.%	388/283	10.1	[95]
Laminated structure	Copper (Cu)	CNT/4 vol%	232/183	34.5	[171]
Trimodal UFG structure	2024 Al	CNT/1.5 wt.%	722/559	7.4	[126]
Bimodal grain structure	2024 Al	CNT/1.5 wt.%	462/434	5.7	[40]
Laminated structure	Al	RGO + Al_2_O_3_/0.15 vol% + 5 vol %	270/203	14.7	[153]
Laminated structure containing a network configuration of SiC	Al	Nano-SiC/10 vol%	340/316	2.5	[43]
UFG structure contaning discontinuous layered structures	6111 Al	Nano-ZrB_2_ + graphene nanoplates (GNP)/0.3 wt.% + 3 wt.%	434/349	13.8	[50]
Inverse nacre structure	Magnesium	Nano-SiC/5 vol%	284/255	15.1	[54]
Heterogeneous lamella structure	Al	Al_2_O_3_/5 wt.%	349/289	26	[172]
Functionally graded materials with three-layered structure	7075 Al	Micron-B_4_C/7.5 vol%	1291/789	18.2	[138]

## Data Availability

Not applicable.

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
