# Peer review of "Reviewing the Integrated Design Approach for Augmenting Strength and Toughness at Macro- and Micro-Scale in High-Performance Advanced Composites"

_materials, 2023, doi:10.3390/ma16175745_

Round 1
Reviewer 1 Report
The present paper includes interesting results regarding the mechanical properties of composites, which can be suitable for journal of MATERIALS. Anyhow, the reviewer would like to make the following comments
1. Authors listed polymer and metal composites in this study to address the problems and solution during the manufacturing or lifetime. However, no data is available for ceramic matrix composites. Therefore, the title must change.
2. Authors reported that the gradient configuration using particle size can improve the mechanical properties particularly toughness and strength. So, it is necessary to mention in what composites materials: what matrix and second phase (fibre, particles...)
3. Heat treatment plays a significant role in influencing the mechanical properties of metal matrix composites. So, I recommend that authors add some sentences to display its importance for composites
4. Achieving a uniform distribution of the second phase in composites is essential for ensuring consistent and predictable material properties. So, the authors must justify, how to get the target (good uniform distribution of second phase)
5. Composites are made up of different materials with varying properties, leading to anisotropy (direction-dependent behavior). This complexity makes it difficult to predict and optimize the response of the structure under different loading conditions. So, I recommend that the authors add a summary of issues in the conclusion e,g Addressing these challenges often requires a combination of advanced modeling techniques, optimization algorithms, experimental validation, and interdisciplinary collaboration. As the field of composite materials continues to evolve, researchers and engineers are constantly working to develop innovative approaches to overcome these challenges and create efficient and reliable composite structures
Author Response
Comments and Suggestions for Authors
The present paper includes interesting results regarding the mechanical properties of composites, which can be suitable for journal of MATERIALS. Anyhow, the reviewer would like to make the following comments
- Authors listed polymer and metal composites in this study to address the problems and solution during the manufacturing or lifetime. However, no data is available for ceramic matrix composites. Therefore, the title must change.
- We appreciate the reviewer's feedback. we intended to focus specifically on the integrated design approach for augmenting strength and toughness in heterogeneous metal matrix composites (MMCs) in this review. To address this concern, we propose the following revised title for our paper:
“Reviewing Integrated Design Strategies for Enhancing Strength and Toughness at Macro and Microscale in High-performance Advanced Composites: Polymer and Metal Matrix Composites”
- Authors reported that the gradient configuration using particle size can improve the mechanical properties particularly toughness and strength. So, it is necessary to mention in what composites materials: what matrix and second phase (fibre, particles...)
- We appreciate the reviewer's feedback. Here is the added paragraph to address the reviewer comment.
“Additionally, in the investigation conducted by Wu et al. [1], a functionally graded AA7075-B4C composite featuring a multi-level gradient structure that encompasses reinforcements, dislocations, and precipitates was examined. The outcomes revealed exceptional bending strength, approximately 1291 MPa, under tensile loading of AA7075, attributed to robust interlayer bonding and the presence of multi-level gradient structures. Furthermore, this multi-level gradient B4C/Al composite showcased high strength without significant interlayer cracks or microstructure deterioration post-deformation, contributing significantly to its overall toughness. Li et al. [2] reported the strain hardening of gradient IF steels by incorporating GNDs that are associated with the non-uniform deformation as experimentally observed on the lateral surface. Their results demonstrated that the strength-ductility performance of gradient samples is remarkably enhanced as compared with their homogeneous counterparts. Furthermore, titanium matrix composites reinforced with continuous carbon fibers showed the correlation between gradient configurations in fiber distribution and the resulting improvements in mechanical properties [3].”
- Heat treatment plays a significant role in influencing the mechanical properties of metal matrix composites. So, I recommend that authors add some sentences to display its importance for composites
- We appreciate the insightful comment regarding the substantial impact of heat treatment on the mechanical properties of metal matrix composites. We added the below paragraph.
“In fact, Heat treatment, as a critical process in materials engineering, significantly influences the microstructure and subsequently the performance of these composites. It can lead to profound changes in characteristics such as strength, ductility, hardness, and even fracture toughness. In the context of this review, it should be stated that the paramount importance of heat treatment is a tool for tailoring the mechanical behavior of heterogeneous metal matrix composites. By subjecting these materials to specific thermal cycles, it becomes possible to manipulate the grain size, distribution of phases, dislocation densities, and even the presence of residual stresses. These modifications often result in enhanced properties, such as improved strength-ductility trade-offs or heightened resistance to fatigue. For instance, in the work of Bingzhu et al. [4], the heat treatment of a titanium-based composite led to a refined microstructure and a subsequent increase in tensile strength due to the dissolution and precipitation of certain phases. Similarly, the investigations by Myriounis and co-authors [5] on aluminum matrix composites revealed that heat treatment played a pivotal role in optimizing the interfacial bonding between the matrix and reinforcement, thereby enhancing overall mechanical properties. “.
- Achieving a uniform distribution of the second phase in composites is essential for ensuring consistent and predictable material properties. So, the authors must justify, how to get the target (good uniform distribution of second phase).
- We sincerely appreciate the reviewer's emphasis on the importance of attaining a uniform distribution of the second phase within composites to ensure consistent and predictable material properties. Achieving this desired uniformity is pivotal as it directly impacts the mechanical, thermal, and other functional characteristics of the composite. To answer the reviewer's comment, we added the below paragraph to the manuscript.
Generally, achieving the uniform dispersion of nanoparticles is a vital concern of making nanoparticles reinforced composites as it directly impacts the mechanical, thermal, and other functional characteristics of the composite. Here is a list of some common approaches used to ensure homogeneous dispersion of the second phase within the matrix:
- Mechanical Mixing: Utilizing mechanical methods such as ball milling, attrition milling, and high-energy mixing can help break down agglomerates and promote a more even distribution of the second-phase particles within the matrix.
- Ultrasonic Dispersion: Ultrasonic waves can be employed to disperse and break down clusters of second-phase particles, ensuring their uniform distribution within the matrix material.
- Powder Metallurgy (PM) Techniques: Powder metallurgy methods, such as hot isostatic pressing (HIP) and spark plasma sintering (SPS), involve consolidating powders under controlled conditions. These methods can result in a more uniform distribution of the second phase due to the homogenizing effect of powder compaction [6].
- Liquid Phase Infiltration: Infiltrating a liquid matrix into a preform of reinforcing particles can help achieve a uniform distribution of the second phase, especially in cases where the reinforcing phase is in fiber or porous form.
- Gradient Precipitation Techniques: Controlling the cooling rate during solidification can promote uniform precipitation of the second phase, resulting in a more homogeneous distribution.
- Surface Treatment and Coating: Treating or coating the second-phase particles before incorporation can improve their wetting and dispersion within the matrix material.
- Electrostatic Dispersion: Employing electrostatic forces can assist in dispersing and separating particles, enhancing their uniform distribution within the matrix.
- Additive Manufacturing: Techniques like 3D printing allow precise control over the placement of the second-phase material, enabling the creation of complex geometries with uniform distribution.
- Melt Infiltration: For fiber-reinforced composites, melt infiltration involves impregnating a porous preform with molten matrix material. This can facilitate a more even distribution of the matrix around the fibers.
- Controlled Agitation and Stirring: Utilizing controlled agitation or stirring during the mixing and consolidation processes can prevent settling and promote uniform distribution.
- Sol-Gel Techniques: Sol-gel processes can result in the formation of a homogenous precursor material that can be used to achieve uniform distribution upon consolidation.
- Smart powder processing: In this method, the process involves utilizing four distinct mechanical devices to induce mechanical forces, leading to the covering of fine shell (guest) particles onto the surface of larger core (host) particles, which in turn develop the advanced production of materials [7,8].
These ideas may be based on the specific type of composite, the composition of the secondary material, and the desired end product. A combination of these methods can also be used to obtain a good view of the second phase in the matrix material.
- Composites are made up of different materials with varying properties, leading to anisotropy (direction-dependent behavior). This complexity makes it difficult to predict and optimize the response of the structure under different loading conditions. So, I recommend that the authors add a summary of issues in the conclusion e,g Addressing these challenges often requires a combination of advanced modeling techniques, optimization algorithms, experimental validation, and interdisciplinary collaboration. As the field of composite materials continues to evolve, researchers and engineers are constantly working to develop innovative approaches to overcome these challenges and create efficient and reliable composite structures.
- We deeply appreciate the reviewer's insight into the intricacies posed by the heterogeneous nature of composite materials, leading to anisotropic behavior and complex response under varying loading conditions. This consideration is indeed essential, and we fully acknowledge the importance of addressing these challenges in our paper's conclusion.
“In composites, the integration of different materials with different materials leads to complex materials that exhibit anisotropic behavior that leads to responses under different loading conditions. This inconsistency poses problems in the estimation and optimization of mixed models. Solving these problems requires an integrated and multifaceted approach. Best modeling methods that take anisotropic behavior into account play an important role in simulating interactions in composites. In addition, the use of optimization algorithms helps to solve the best configuration for a given load scenario, enabling the product to achieve superior performance. Practical tests are essential to bridge the gap between theory and reality. The validation not only validated the prediction of the model's power but also showed subtle interactions between features that might escape theoretical predictions. But the quest to overcome these problems continues beyond discipline. Interdisciplinary collaboration combines the power of information science, mechanics, and engineering to provide new solutions to problems. By leveraging a variety of skills and perspectives, scientists and designers can collectively unlock new insights and ideas to unravel the complexities of composites. As the connected devices market continues to expand, the work of scientists and engineers continues to innovate. This process aims not only to eliminate the problems caused by anisotropy and various properties, but also to create a good and reliable mix to meet the needs of different enterprises.“

Reviewer 2 Report
This paper is very important and interesting, but some problems should be solved:
(1) The abstract should be improved.
(2) The toughness of metal matrix composites reinforced by CNT or graphene is relatively decrease in general, the mechanisms for toughness increased need to be further improved.
(3) How to figure out the energy dissipation during the process of designing and fabricating composite configurations?
(4) The importance of composite configurations should be further explained.
Good!
Author Response
Reviewer #2:
Comments and Suggestions for Authors
This paper is very important and interesting, but some problems should be solved:
- The abstract should be improved.
- Thank you for your valuable comment. I add some useful sentence into abstract.
- The toughness of metal matrix composites reinforced by CNT or graphene is relatively decrease in general, the mechanisms for toughness increased need to be further improved.
- We deeply appreciate the reviewer's insight into the mechanisms for toughness. In answer to the reviewer’s comment, the following paragraph was added to the manuscript.
“Incorporating a gradient distribution of reinforcements within the matrix can lead to a substantial enhancement in both strength and toughness. By varying the concentration or size of reinforcements along specific directions, the initiation and propagation of cracks can be directed away from critical regions. This redirection of crack paths limits catastrophic failure, allowing the material to sustain higher loads and absorb greater energy before fracture occurs. An example of the toughening process in the scientific literature is the use of toughening, which refers to the process of making the material resistant to cracking [9]. When a crack propagates, the associated irreversible work in different materials classes is different. Therefore, the best toughening mechanism varies between different materials [9,10]. Basically, as mentioned earlier, Intrinsic toughening mechanisms are processes that act ahead of the crack tip to increase the material’s toughness. These will tend to be related to the structure and bonding of the base material, as well as microstructural features and additives to it [11,12]. Another example is the strain distribution around a matrix crack bridged by reinforcing members and its effect on tensile fracture [13]. The results show that the strain propagated in the elliptical region around cracks can be controlled by varying the size and height of the reinforcement fibers. This allowed them to redirect crack paths away from critical regions, leading to enhanced strength and toughness [13]. These are some examples of how the distribution gradient of reinforcement can be added to the matrix and increase strength and toughness. There are many other mechanisms and techniques that can be used to achieve this goal.
Considering the toughening of microheterogeneous configurations of reinforcements, it is evident that refining the interaction between the matrix and the reinforcing phase can yield remarkable improvements in toughness for CNT- or graphene-reinforced MMCs. Yizhang Liu et al. [14], discussed the developmental stage of ceramic matrix composites strengthened with nanocarbon materials. The study explored the influence of interface bonding conditions on the fundamental properties of these composites and examined strategies to enhance this interface bonding. Afifah Md Ali et al. [15], three key topics were covered concerning the challenges associated with employing graphene as a reinforcement material. These subjects encompassed the mechanisms through which graphene enhanced the strength of metal matrices and the limitations that restricted the extent of property improvement. Surface modification techniques applied to the reinforcing phase can optimize interfacial bonding. By functionalizing the surfaces of CNTs or graphene with suitable chemical groups, compatibility with the matrix can be enhanced, ensuring robust load transfer and preventing premature debonding. Rama Dubey et al. [16] , investigated how modifying the surfaces of carbon nanotubes (CNTs) could optimize their interaction with other materials. By introducing specific chemical groups onto CNT surfaces, compatibility with surrounding matrices was heightened, resulting in robust load transfer and the prevention of premature detachment. Additionally, engineering hybrid structures where CNTs or graphene are coupled with secondary reinforcements, such as nanoparticles or fibers, can impart multifaceted toughening mechanisms. This synergistic combination of reinforcements introduces mechanisms like crack deflection by secondary phases, crack pinning by nanoparticles, and stress redistribution through fiber bridging. The resulting interactions mitigate crack propagation and contribute to the overall enhancement of toughness. Jyoti et al. [17], explored the realm of hybrid structures where graphene or carbon nanotubes were combined with supplementary reinforcements like nanoparticles or fibers. This integration gave rise to diverse toughening mechanisms, including the deflection of cracks by secondary phases, immobilization of cracks by nanoparticles, and redistribution of stress via fiber bridging. The collaborative effects of these mechanisms hindered crack propagation and collectively enhanced the overall toughness of the composite materials.
It is important to note that both macro- and micro-scale configurations of reinforcements significantly influence the balance between toughness and strength in metal composites. However, achieving a satisfactory spatial dispersion of reinforcements requires careful consideration. For example, achieving a uniform intragranular dispersion of CNTs in an aluminum matrix necessitates specific conditions related to time, energy, external applied stress, and thermal activation [18,19]. Moreover, challenges remain in overcoming the strength-toughness trade-off, emphasizing the need for further extensive research in the field of strengthening and toughness mechanisms. In summary, addressing the relative decrease in toughness observed in CNT- or graphene-reinforced MMCs requires a dual strategy encompassing both macro and micro heterogeneous configurations of reinforcements. Through strategic design choices, such as gradient distributions, sacrificial phases, surface functionalization, and hybrid structures, the mechanisms for toughness enhancement can be harnessed to significantly improve the overall performance of these advanced composite materials.
- How to figure out the energy dissipation during the process of designing and fabricating composite configurations?
- We appreciate the reviewer's interest in understanding how energy dissipation is considered during the design and fabrication of composite configurations. The below paragraph was added to the manuscript.
Energy dissipation during the creation of composite configurations is closely related to the material properties, mechanical forces, and intrinsic properties of the composite materials.
Advanced modeling techniques are essential to measure the removal of this force. Numerical simulations, such as finite element analysis (FEA) and molecular dynamics (MD), provide a virtual platform for predicting the behavior of designed configurations under load conditions. These simulations provide insights into how energy is absorbed and localized within the composite, contributing to enhanced fracture toughness [143]. Furthermore, experimental validation helps quantify the energy dissipation. Techniques such as mechanical testing, dynamic impact testing, and acoustic emission analysis provide empirical evidence of energy dissipation during loading. By connecting experimental results to theoretical predictions, it enriches the understanding of energy dissipation mechanisms is enriched, allowing for the refinement of design configurations to optimize toughness [144,145]. The new research focuses on uncovering the scientific principles that govern energy dissipation in composite configurations. From this finding, the “how and why” of energy dissipation mechanisms leads to the development of general theories to guide design choices. The combination of theoretical insight, numerical simulation, and experimental validation empowers researchers to quantify and optimize energy dissipation mechanisms within composite configurations [146]. More importantly, the influence of microstructural components such as interfaces, dislocations, cracks, and local plastic deformation fields on energy dissipation cannot be underestimated. It is important to fine-tune the microstructures at different scales. Advances in nanotechnology and materials science have made it possible to combine macroscopic and nanoscale levels to enable optimization of energy dissipation mechanisms [147,148]. However, it is recognized that the development of multi-stage processes and multiple models presents challenges with “shape control” and “controllability.” Achieving controlled fabrication necessitates a deep understanding of materials processing techniques, innovative manufacturing methodologies, and collaborative interdisciplinary efforts [149].
- The importance of composite configurations should be further explained.
- Thank you for your valuable comment. I add the below paragraph in the manuscript.
“Composite configurations serve as the blueprint for designing advanced materials beyond the limitations of individual components. These configurations are strategically designed arrangements of different phases, materials, and microstructures integrated to achieve specific mechanical, electrical, thermal, and functional properties. The importance of composite configurations lies in their profound influence on material behavior, enabling the creation of high-performance materials that can meet the demands of modern engineering applications. Composite configurations can be customized to meet different requirements. By intelligently designing the arrangement of constituent phases, scientists can select the desired properties while mitigating limitations. For example, the incorporation of reinforcing phases in MMCs can drastically enhance stiffness and strength, empowering the material to withstand demanding mechanical loads. Similarly, the incorporation of thermally conductive phases into polymer-based composites can contribute to good thermal conductivity, which is important for applications such as electricity. Moreover, composite configurations enable the manipulation of mechanical performance attributes such as toughness, fatigue resistance, and fracture behavior. By combining these methods, researchers can arrest crack propagation, deflect crack paths, and enhance energy dissipation mechanisms, thereby enhancing fracture toughness and overall mechanical integrity. This design freedom also extends to tailoring anisotropic behavior, where material properties are varied along different directions, allowing optimum operation for certain conditions. Furthermore, composite configurations combine different components into a single material system, enabling materials to excel in multiple roles simultaneously. Beyond performance benefits, composite configurations contribute to sustainable design principles by reducing material consumption and weight, thus minimizing environmental impacts. In essence, composite configurations represent the forefront of materials engineering; composites provide unprecedented possibilities, performance and stability through the use of microstructure and interfaces.“

Reviewer 3 Report
The present review deals with the problem of simultaneously achieving strength and toughness in metal matrix composites. It will be useful to the readers, inluding both reseachers and students. I have several comments on the paper, please see below.
1. It seems that this paper would have benefited from the addition of more illustrations. Also, addiing tables and schemes summarizing the content is very much encouraged.
2. If a figure is re-used from another source, the permission should be obtained and the caption of the figure should contain the following information: "Reprinted from [YYY], with permission from XXX". If the source was published open access, this should also be stated in the caption with a reference to the pertaining license.
3. For me, the term "heterogeneous metal matrix composites" is unclear. It is recommended to use another term. All composites are heterogeneous, as there are interfaces between the phases.
4. "... a homogeneous distribution of reinforcements": the term "uniform" suits better here.
5. Author contributions are not provided.
6. The Summary is not informative in the present state. It is too general by the statements provided. I think the essence of the composite design approaches discussed in the review should be briefly given here.
The quality of the text in terms of the language is fine.
Author Response
Reviewer #3:
The present review deals with the problem of simultaneously achieving strength and toughness in metal matrix composites. It will be useful to the readers, inluding both reseachers and students. I have several comments on the paper, please see below.
- It seems that this paper would have benefited from the addition of more illustrations. Also, addiing tables and schemes summarizing the content is very much encouraged.
- We appreciate the reviewer's feedback and their suggestion to include more illustrations in the paper. While we recognize the potential benefits of incorporating additional figures and tables, we would like to clarify the specific intent and scope of this review paper. The primary objective of this paper is to provide a critical and concise overview of the strategies for enhancing strength and toughness in composite materials. As such, our focus has been on presenting the core concepts and key findings within a limited space. It is important to note that this review paper is intended to be succinct, highlighting fundamental aspects of the discussed strategies. To maintain its concise nature, we have prioritized the clarity and precision of our explanations, and we have minimized the use of supplementary illustrations. We believe that this approach best serves the purpose of conveying essential information efficiently within the confines of a short review paper. However, we genuinely value the reviewer's suggestion, and we are taking it into consideration for our future endeavors. Our upcoming long-form review paper will allow us to provide more comprehensive coverage and dedicate space to a greater number of figures, tables, and visual aids. This will enable us to elucidate complex concepts in a more visual and accessible manner, enhancing the reader's understanding.
Once again, we appreciate the reviewer's insights and assure them that their feedback will influence our approach in future review papers.
- Regarding the second part of your comment, the below paragraph was added to the manuscript.
In the beginning, a concise summaries for each of the sections are provided as follow:
- Introduction:
The introduction sets the stage by highlighting the significance of composite materials in modern engineering. It outlines the challenges posed by conflicting material properties and emphasizes the need for innovative approaches. The introduction also introduces the concept of heterogeneous configurations and previews the subsequent sections where different strategies for enhancing strength and toughness will be discussed.
- Strengthening and Toughening through Macro Heterogeneous Configuration of Reinforcement:
This section explores the concept of utilizing macro heterogeneous reinforcement arrangements to enhance strength and toughness in composite materials. It discusses gradient distributions of reinforcement, sacrificial phases, and their effects on crack propagation and energy dissipation. The section highlights how these design approaches mitigate limitations and optimize material behavior.
- Strengthening and Toughening of Microheterogeneous Configurations of Reinforcements:
Focusing on microheterogeneous configurations, this section delves into techniques for improving toughness through refined interactions between matrix and reinforcement. It emphasizes surface modifications, hybrid structures, and the role of microstructure in toughening mechanisms. The discussion highlights how tailored combinations of reinforcement phases can lead to enhanced mechanical performance.
- Strengthening and Toughening the Matrix through Intrinsic Mechanisms:
This section examines intrinsic mechanisms within the matrix that contribute to enhanced strength and toughness. It explores processes such as grain refinement, dislocation interaction, and precipitation hardening. The section underscores how these intrinsic mechanisms are harnessed to improve mechanical properties.
- Designing and Fabricating Composite Configurations based on Energy Dissipation:
Focusing on energy dissipation, this section delves into the role of design configurations in enhancing fracture toughness. It explains the significance of preventing cracking and maximizing energy dissipation from both static plastic deformation and crack propagation. The section also explores the importance of micro and nano scale heterostructures in controlling crack behavior and optimizing energy dissipation.
- Summary and Outlook:
In the summary section (6.1), the paper recaps key findings from the discussed strategies for enhancing strength and toughness in composite materials. It emphasizes the potential of innovative research to uncover scientific principles driving improved properties. In the outlook section (6.2), the focus shifts to future directions, advocating for the development of energy dissipation theories and optimization of composite configurations. The importance of interdisciplinary collaboration and learning from biological systems is also highlighted.
- If a figure is re-used from another source, the permission should be obtained and the caption of the figure should contain the following information: "Reprinted from [YYY], with permission from XXX". If the source was published open access, this should also be stated in the caption with a reference to the pertaining license.
- We appreciate the reviewer's attention to the proper handling of figures sourced from external works. As part of our commitment to adherence with copyright and attribution standards, we acknowledge the necessity of obtaining permissions for reused figures and providing accurate information in the caption.
- For me, the term "heterogeneous metal matrix composites" is unclear. It is recommended to use another term. All composites are heterogeneous, as there are interfaces between the phases.
- We appreciate your valuable feedback regarding the discussion on heterogeneous materials. However, we respectfully hold a differing perspective from the esteemed reviewer. It's important to emphasize that the heterogeneity present in the microstructure of these materials significantly diverges from the traditional notion of heterogeneity found in conventional composites. This unique form of heterogeneity is deliberately generated through meticulous design and customization processes involving the initial materials, or processing. In essence, researchers are actively employing a bottom-up approach to precisely engineer and create a distinct microstructure, often referred to as an "architecture." The intention is to harness this approach for tailoring specific properties within the material. Notably, there has been a surge in recent publications dedicated to exploring and elucidating the characteristics of such materials. I've incorporated references to some of these pertinent publications in my work.
The below paragraph was also added to the manuscript.
Indeed, this aligns with the suggested characterization of heterogeneous materials, which are composed of regions exhibiting significant variations in strength. These regions can vary in size from micrometers to millimeters. To be more exact, heterogeneous materials represent a novel category of substances that exhibit exceptional combinations of strength and ductility beyond the reach of their homogeneous equivalents [20-23]“.
- "... a homogeneous distribution of reinforcements": the term "uniform" suits better here.
- Thank you for your precise attention. The comment was applied.
- Author contributions are not provided.
- Author contributions were provided.
- The Summary is not informative in the present state. It is too general by the statements provided. I think the essence of the composite design approaches discussed in the review should be briefly given here.
Thank you for your precise attention. The conclusion was completely revised as follows.
“In summary, this review article explores the structural design of MMCs, with particular emphasis on the varying role of heterogeneous nanostructured metals in improving strength and toughness. The main level of decoration, size, and variety of non-uniform structures are important driving factors when analyzing the evolution of the configuration of MMCs. This explanation leads to the future of theory: exploring energy distribution and deformation from a new perspective beyond the localization constraints.
At the same time, the field of quantitative analysis gained importance, providing a complete picture of the strength and toughness mechanisms within composite materials. Expressing the interaction of energy between scales is meaningful, showing energy dissipation between scales as a determinant of crack initiation and propagation. This understanding paves the way for changes in structural dynamics, material strength, and behavior while orchestrating inverse design trajectories for multiphase, multiscale reinforced phase, and matrix composite configurations/interfaces. In the continuation of this development, the principle of combining technology based on the configuration of various components takes the lead. This principle, which led to the careful design of composite configurations/interfaces at micro and macro scales, must overcome the well-established strength-toughness reversal dilemma in metal matrix composites. The culmination arrives with a breakthrough in the stress-strain distribution at the same scale and promises to break the bottleneck that hinders the visualization of energy and synergy. This document crosses the path from the foundations to the future through the lens of innovation, highlighting the potential for deformation in the interaction of settings in MMC.
In composites, the integration of different materials with different materials leads to complex materials that exhibit anisotropic behavior that leads to responses under different loading conditions. This inconsistency poses problems in the estimation and optimization of mixed models. Solving these problems requires an integrated and multifaceted approach. Best modeling methods that take anisotropic behavior into account play an important role in simulating interactions in composites. In addition, the use of optimization algorithms helps to solve the best configuration for a given load scenario, enabling the product to achieve superior performance. Practical tests are essential to bridge the gap between theory and reality. The validation not only validated the prediction of the model's power but also showed subtle interactions between features that might escape theoretical predictions. But the quest to overcome these problems continues beyond discipline. Interdisciplinary collaboration combines the power of information science, mechanics, and engineering to provide new solutions to problems. By leveraging a variety of skills and perspectives, scientists and designers can collectively unlock new insights and ideas to unravel the complexities of composites. As the connected devices market continues to expand, the work of scientists and engineers continues to innovate. This process aims not only to eliminate the problems caused by anisotropy and various properties, but also to create a good and reliable mix to meet the needs of different enterprises.“

Round 2
Reviewer 3 Report
The title of the paper has been changed and now it is unsuitable. Why polymers are included in the title? As the Authors stated, their focus was MMCs. The term "metal matrix composites" is now for some reason repeated twice, which is unacceptable for a paper title.
The newly added text should be carefully proofread. For example, the phrase "the integration of different materials with different materials ...", "The validation not only validated ..." should be revised. What is "the main level of decoration"?
The problem of the text is wordiness and structure of sentences, which is in many places too complex for a technical paper.
The format of the revised manuscript does not fit the Journal template and it is difficult to read the paper and consider the changes made during the revision. The initial manuscript was prepared in the template.
I still believe that a table with examples of successfully designed composites and their mechanical properties would have been a good addition to the review.
It seems that the revised version was prepared in a big hurry. Proofreading is needed.
The problem of the added text is wordiness and too complex a structure of some sentences.
Author Response
Dr. Behzad Sadeghi
University del Salento, Lecce, Italy.
Email address: B.sadeghi2020@gmail.com
Dear Reviewers,
We would like to express our sincere gratitude for your thorough review of our manuscript entitled "Reviewing the Integrated Design Approach for Augmenting Strength and Toughness at Macro and Microscale in Heterogeneous Metal Matrix Composites " submitted to Materials. We appreciate the time and effort you have dedicated to evaluating our work and providing valuable feedback.
We have carefully considered your comments and suggestions, and we have made several revisions to address the concerns raised. In response to each of your specific points, we provide our detailed explanations and the corresponding changes made in the revised manuscript in green highlight, and reviwer respond in blue.
Reviewer #3:
- The title of the paper has been changed and now it is unsuitable. Why polymers are included in the title? As the Authors stated, their focus was MMCs. The term "metal matrix composites" is now for some reason repeated twice, which is unacceptable for a paper title.
- I changed the title with “Reviewing the Integrated Design Approach for Augmenting Strength and Toughness at Macro and Microscale in High-performance Advanced Composites”
- The newly added text should be carefully proofread. For example, the phrase "the integration of different materials with different materials ...", "The validation not only validated ..." should be revised. What is "the main level of decoration"?
- All the added text was carefully checked and the mentioned concern was solved.
- The problem of the text is wordiness and structure of sentences, which is in many places too complex for a technical paper.
- All the added text was carefully checked and the mentioned concern was solved.
- The format of the revised manuscript does not fit the Journal template and it is difficult to read the paper and consider the changes made during the revision. The initial manuscript was prepared in the template.
- The format issue of the journal was solved.
- I still believe that a table with examples of successfully designed composites and their mechanical properties would have been a good addition to the review.
- Table 1 was added to the manuscript with its corresponding explanation.
Comments on the Quality of English Language
- It seems that the revised version was prepared in a big hurry. Proofreading is needed.
- Proofreading was done.
- The problem of the added text is wordiness and too complex a structure of some sentences.
- We worked on simplifying the language and structure of certain sentences in the added text to reduce wordiness and improve readability. Indeed, We almost worked on the whole of the text.

Round 3
Reviewer 3 Report
The paper has been revised according to the Reviewer's suggestions.